# Feasibility of reporting results of large randomised controlled trials to participants: experience from the Fluoxetine Or Control Under Supervision (FOCUS) trial

Gillian Mead [iD],[1] Martin Dennis,[2] FOCUS trial Collaboration

[1]Geriatric Medicine, University of Edinburgh, Edinburgh, UK
[2]Centre for Clinical Brain Sciences, University of Edinburgh, Edinburgh, UK

**Correspondence to**
Professor Gillian Mead;
gillian.e.mead@ed.ac.uk

## ABSTRACT

**Objectives** Informing research participants of the results of studies in which they took part is viewed as an ethical imperative. However, there is little guidance in the literature about how to do this. The Fluoxetine Or Control Under Supervision trial randomised 3127 patients with a recent acute stroke to 6 months of fluoxetine or placebo and was published in *the Lancet* on 5 December 2018. The trial team decided to inform the participants of the results at exactly the same time as *the Lancet* publication, and also whether they had been allocated fluoxetine or placebo. In this report, we describe how we informed participants of the results.

**Design** In the 6-month and 12-month follow-up questionnaires, we invited participants to provide an email address if they wished to be informed of the results of the trial. We re-opened our trial telephone helpline between 5 December 2018 and 31 March 2019.

**Setting** UK stroke services.

**Participants** 3127 participants were randomised. 2847 returned 6-month follow-up forms and 2703 returned 12-month follow-up forms; the remaining participants had died (380), withdrawn consent or did not respond.

**Results** Of those returning follow-up questionnaires, a total of 1845 email addresses were provided and a further 50 people requested results to be sent by post. Results were sent to all email and postal addresses provided; 309 emails were returned unrecognised. Seventeen people replied, of whom three called the helpline and the rest responded by email.

**Conclusion** It is feasible to disseminate results of large trials to research participants, though only around 60% of those randomised wanted to receive the results. The system we developed was efficient and required very little resource, and could be replicated by trialists in the future.

**Trial registration number** ISRCTN83290762; Post-results.

## Strengths and limitations of this study

► This is the first large trial to report disseminating results to participants, including treatment allocation.
► We demonstrated the feasibility of our approach, which was approved by regulatory authorities.
► However, using email to disseminate results meant that some emails were returned unrecognised.

editorial in the *British Medical Journal* stated that the results of clinical trials should be disseminated to those who took part in them because this is a courteous and an ethical imperative.[2] The World Medical Association's Declaration of Helsinki states that 'all medical research participants should be given the opportunity of being informed about the general outcome and results of the study'. The National Institute for Health Research states that it is important to establish whether a participant will want to be actively informed of trial results, or whether they would like the onus to be left with them to obtain the results.[3]

Contacting participants many years after enrolment might be an upsetting reminder of their illness, though one small study in cancer suggested that informing participants might increase their understanding of the trial results.[4] Furthermore, some might argue that trying to contact participants who have died might not be ethical. To our knowledge, there is, however, no practical guidance in the literature about the steps required to inform participants of the results of the trials in which they participated. Furthermore, it is not known whether research participants do wish to receive results of the trial, which can sometimes be many years after they had been enrolled.

## INTRODUCTION

Eighteen years ago, an article published in the *Journal of the American Medical Association* recommended that participants be informed of the results of the clinical trials in which they participate.[1] In November 2019, an

BMJ

In this brief report, we describe how we disseminated the results of a large multicentre randomised controlled trial, Fluoxetine Or Control Under Supervision (FOCUS), and individuals' treatment allocation and the feedback received from participants.

## METHODS

FOCUS was a pragmatic, multicentre, parallel group, double-blind, randomised, placebo-controlled trial done at 103 hospitals in the UK and recruiting 3127 patients between 10 September 2012 and 31 March 2017, testing whether a 6-month course of fluoxetine given 2–15 days after stroke would improve recovery at 6 months.[5]

### Patient and public involvement

We involved a patient and public involvement (PPI) group of stroke survivors and carers during its design. The group recommended disseminating results to participants who had taken part; this included the family members of participants who had died; the rationale being that family members might be interested in results of the trials in which their loved one had participated. Our grant application to National Institute for Health Research Health Technology Assessment included a request for funding for dissemination of results and this was approved.

Subsequently, two lay members identified through the UK Stroke Research Network who had not been involved in the initial planning stages were invited to sit on the trial steering committee, and advised on how to disseminate results to participants.

### Methods of trial follow-up

All the trial follow-ups were by postal questionnaires at 6 and 12 months; participants who did not respond were contacted by telephone and the questionnaires completed over the telephone. The postal questionnaire at both 6 and 12 months concluded with 'If you want to find out more about the trial and its results (in about 2018) please enter an email address where we can contact you, or a person close to you'. We did not record whether the email address provided was for the participant or for someone else (eg, a family member). We did not collect the email address at the time of recruitment.

During follow-up telephone calls with participants who had not returned their 6-month or 12-month follow-up questionnaires, we noted that many participants asked us what their treatment allocation had been. We could not provide this information at the time of the call because we had to remain blinded to treatment allocation, but we decided to include individual treatment allocation when we contacted participants with the results of the trial. To the best of our knowledge, this has never been done before.

We considered whether to disseminate the results to all participants by post; although we had costed for the postage in the grant application (£1 per letter), we realised that this would have required substantial resource, including research staff time (estimated 5 min per letter, which is about 125 hours for 1500 letters, plus paper as well as the postage), and if participants had moved house, confidentiality might have been compromised. Thus, we decided to use whatever email had been provided in the follow-up questionnaires, and only use post if there was no email address.

The content of the email to participants was drafted by the trial team and edited by the two PPI members of the trial steering committee. The email was written in lay language, starting with an explanation of why we were contacting them and thanking them again for having taken part, provided their treatment allocation and the overall results of the trial (online supplemental materials), links to further information and the number for the telephone helpline. We did not check whether participants were still alive.

We sent the email at exactly the same time as *the Lancet* publication on 5 December 2018[5] and the presentation of the main results at the UK Stroke Forum. We managed to coordinate the emails, the presentation and the publication through close liaison with *the Lancet* and the UK Stroke Forum scientific committee. At the same time, we also re-opened the trial telephone helpline (Monday to Friday, 09:00 to 17:00) until 31 March 2019, so that participants receiving the email could contact us if they had any questions. We recorded how many responses were received and the reason for their response. A co-principal investigator (MD) then responded to the participant or family member if this seemed appropriate. We did not follow-up emails that were returned as unrecognised by the mail subsystems.

## RESULTS

Of the 3127 participants randomised, 2847 returned 6-month follow-up forms and 2703 returned 12-month follow-up forms; the remaining participants had died (380), withdrawn consent or did not respond. Of the returned forms, 1845 email addresses were provided and a further 50 people requested results by post. Thus, the number who wished to be informed of results was 1895; this represents 60% of all participants and 70% of those who could be contacted at 12 months.

We therefore sent emails containing the results of FOCUS and treatment allocation from the FOCUS email address (focus.trial@ed.ac.uk) to 1845 email addresses, and a paper letter to the 50 postal addresses.

Three hundred and nine emails were returned as unrecognised by the email subsystems.

Seventeen people (0.9%) (seven participants, eight family members and two unknown) replied; of these, three called the helpline and the rest replied by email. Replies were all received between 5 December 2018 and 18 December 2019. Seven expressed thanks for letting them know, two asked for advice on how to read the information we had sent them, and the rest reported on their current health status or informed us of the death

of the participant. A co-principal investigator (MD) telephoned three participants back and emailed the rest of the respondents to thank them for contacting us, offered condolences to the bereaved relatives or provided further information that had been requested.

## DISCUSSION

We have demonstrated that it is feasible to inform trial participants of the results of a large, pragmatic clinical trial and also their treatment allocation. This required planning, involvement of PPI representatives, coordination of the email dissemination with the publication of the trial in *the Lancet* and approval from the Research Ethics Committee, but surprisingly little resource in terms of research staff time and consumables.

However, almost half of the participants did not wish to be informed of the results. We did not explore the reasons for not wishing to receive the results. After we had disseminated the results, only a handful contacted us, mostly to thank us for letting them know. All the responses were within a few days of receiving the results. Only three people called the helpline—this is far lower than the number who typically contacted the hotline during the trial, which had been about four per week. For future trials, we would probably not reopen the helpline and just provide an email contact address for any queries.

To the best of our knowledge, this is the first large randomised trial to report experiences of informing research participants of the results of the trial and also each person's treatment allocation. We did not try to contact participants by post if the email had 'bounced'; this was for practical reasons of resources and cost. We did not record whether the email address provided for receipt of results was for the patient or for a family member, and so we cannot report how many results were received by the participants themselves or by a family member.

Are there any ethical problems with informing participants? Our first participants were enrolled in 2012, and the results were sent to them more than 6 years later. It is possible that participants might have been upset to be reminded of their stroke so long after their enrolment. However, although FOCUS was not designed to explore this, we found no evidence that receiving results was distressing. In theory, families might have been upset to receive an email had their loved one died, but all the bereaved family members who contacted us expressed their appreciation of having been informed. In a review of empirical research about informing participants, the drawbacks might be increased anxiety, anger, guilt or feeling upset, while benefits might include pleasure, satisfaction and relief.[6]

Although we have demonstrated that it is feasible to contact research participants by email, we did not formally explore the thoughts and feelings of research participants when they received the results. This would ideally have required a qualitative substudy which was not an aim of our study.

### Implications for practice

Disseminating results of trials to research participants should be viewed as an ethical imperative.[2] The model for dissemination that we developed with our PPI representatives was feasible. We recommend that trialists consider using our approach, that funders provide funding for this, and that ethics committees approve future requests to use this approach.

### Implications for research

Further research is required to explore why some participants do not wish to receive results of studies in which they participated, whether participants generally wish to know individual treatment allocation and how they wish to receive the information (eg, by email, by post or being provided with a link to a website).

**Acknowledgements** Recruitment and follow-up was supported by the National Institute for Health Research-funded UK Stroke Research Network and the Scottish Stroke Research Network, which was supported by NHS Research Scotland (NRS). Scotland A Research Ethics Committee approved the trial (11/SS/0100) on 21 December 2011. We would like to thank all the participants and their families, the Dundee Speakeasy group who devised the accessible patient information leaflets, and the patient and public involvement (PPI) group who advised us on the design of Fluoxetine Or Control Under Supervision and advocated for participants to be informed of results. We would like to thank David Burgess (a PPI member of our group) for helpful comments on this manuscript.

**Collaborators** FOCUS Trial Co-ordinating Centre: R Anderson, D Buchanan, A Deary, J Drever, R Fraser, C Graham, K Innes, C McGill, A.Mcgrath, D Perry, P Walker, C Williams. Telephone 6 and 12 month follow up (? numbers for each?): R Anderson, M Dennis, A Barugh, G Blair?, Y Chun, E Maschauer, G Mead, M Scott, C Williams, others. Writing Group: M Dennis (Chair), J Forbes, C Graham, M Hackett, G Hankey, A House, S Lewis, E Lundström, G Mead. Trial Steering Committee: Stroke Association funded phase – Peter Sandercock (Chair), Steff Lewis (Statistician), Judith Williamson (Patient involvement), Martin Dennis (Co CI), Gillian Mead (Co-CI), John Forbes (Health Economist), Graeme Hankey (AFFINITY), Maree Hackett (AFFINITY), Veronica Murray (EFFECTS), Karen Innes (Trial Manager), Ray French (Sponsor representative). NIHR funded phase -David Stott (Chair), David Burgess (Lay member), Jonathan Emberson (Independent statistician), Graham Ellis, Pippa Tyrrell, Judith Williamson (Patient involvement), Martin Dennis (Co CI), Gillian Mead (Co-CI), Karen Innes (Trial Manager), Ray French (Sponsor representative). Co-applicants on funding applications. Stroke Association: G Mead, M Dennis, P Sandercock, M MacLeod, S Lewis, F Sullivan, A House, J Forbes, M Hackett, C Anderson, G Hankey. NIHR: G Mead, M Dennis, P Sandercock, M MacLeod, S Lewis, D Morales, A House, J Forbes, M Hackett, C Anderson, G Hankey. Independent Data Monitoring Committee: P Langhorne, F Reid, H Rodgers. Updating systematic review of randomised controlled trials of Fluoxetine in stroke: G Mead. Investigational Medicinal Product: Manufactured by Unichem (Goa). Sourced through – Niche Generics and Discovery Pharmaceuticals. Packaged and distributed by Bilcare, then Sharp. We would like to acknowledge all the patients and their families who participated in FOCUS, the nursing staff who assisted at collaborating sites and the UK stroke research network staff without whom the trial would not have been possible. Participating centres: We have listed each hospital with the total number of patients recruited in [n], followed by the names of the local principal investigator(s), and other significant contributors in that centrel. The hospitals are ordered depending on the numbers recruited. Recruiting centres: Royal Infirmary Edinburgh, Edinburgh [141] (G Mead (PI), N Hunter, R Parakramawansha, A Fazal, P Taylor, W Rutherford, K McCormick, R Buchan, A MacRaild, Y Chun, R Paulton, S Burgess, D McGowan, J Skwarski, F Proudfoot, R Murphy, A Barugh, J Perry); Leeds General Infirmary, Leeds [123] (J Bamford (PI), C Bedford, D Waugh, E Veraque, M Kambafwile, L Makawa, P Smalley, M Randall, L Idrovo, A Hassan, T Thirugnana-Chandran, R Vowden, J Jackson); St Thomas Hospital, London [115] (A Bhalla (PI), C Tam, Prof. A Rudd, C Gibbs, J Birns, L Lee Carbon, E Cattermole, A Cape, L hurley, K Marks, S Kullane); Royal Hampshire County Hospital, Winchester [110] (N Smyth (PI), E Giallombardo, C Eglinton, J Wilson, D Dellafera, P Reidy, M Pitt, L Sykes, A Frith, V Croome, J Duffy, D Cooke, M Hancevic, L Kerwood, C Narh, C Merritt, J Willson); Royal Hallamshire Hospital, Sheffield [107] (A Ali (PI), S Bell, T Jackson, H Bowler, C Kamara, A Naqvi, J Howe, K Stocks, G Dunn, K Endean, F Claydon, S Duty,

C Doyle, K Harkness, E Richards, M Meegada, A Maatouk, L Barron, K Dakin, R Lindert, Prof. A Majid); Calderdale Royal Hospital, Halifax [81] (P Rana (PI), A Nair, C Brighouse-Johnson, J Greig, M Kyu, S Prasad, M Robinson, B Mclean, I Alam, L Greenhalgh, Z Ahmed); University Hospitals of North Midlands NHS Trust, Stoke-on-Trent [80] (Prof. C Roffe (PI), S Brammer, A Barry, C Beardmore, K Finney, H Maguire, P Hollinshead, J Grocott, I Natarajan, J Chembala, R Sanyal, S Lijko, N Abano, A Remegoso, P Ferdinand, S Stevens, C Stephen, P Whitmore, A Butler, C Causley, R Varquez, G Muddegowda, R Carpio, J Hiden, H Denic); Royal Devon & Exeter Hospital, Exeter [73] (J Sword (PI), F Hall, J Cageao, S Keenan, R Curwen, M James, P Mudd, C Roughan, H Kingwell, A Hemsley, C Lohan, S Davenport, T Chapter, A Bowring, M Hough, D Strain, K Gupwell, K Miller, A Goff, E Cusack, S Todd, R Partridge, G Jennings, K Thorpe, J Stephenson, K Littlewood); Monklands Hospital, Airdrie [70] (M Barber (PI), F Brodie, S Marshall, D Esson, C McInnes, I Coburn, F Ross, V Withers, E Bowie, H Barcroft, L Miller); York Hospital, York [69] (P Willcoxson (PI), M Keeling, M Donninson, R Evans, D Daniel, J Coyle, M Elliott, P Wanklyn, J Wightman, E Iveson, A Porteous, N Dyer, M Haritakis, M Ward, L Wright, J Bell, C Emms, P Wood, P Cottrell, L Doughty, L Carr, C Anazodo, M O Neill, J Westmoreland, R Rodriguez, R Mir, C Donne, E Bamford, P Clark Brown); Pinderfields Hospital, Wakefield [67] (A Stanners (PI), I Ghouri, A Needle, M Eastwood, M Carpenter, P Datta, R Davey, F Razik, G Bateman, J Archer, V Balasubramanian, L Jackson, L Benton, J Ball, R Bowers, J Ellam, K Norton); Southend University Hospital NHS Foundation Trust, Essex [64] (P Guyler (PI), S Tysoe, P Harman, A Kundu, T Dowling, S Chandler, O Omodunbi, T Loganathan, S Noor, S Kunhunny, D Sinha, A Siddiqui, A Siddiqui, M Sheppard, S Shah, S Kelavkar, K Ng, L Wilson, A Ropun, L Kamuriwo, R Orath Prabakaran, E France, S Rashmi); Pilgrim Hospital, Boston [63] (D Mangion (PI), C Constantin, S Markova, A Hardwick, J Borley, L De Michele Hock, T Lawrence, J Fletcher, K Netherton, R Spencer, H Palmer); Lincoln County Hospital, Lincoln [60] (M Soliman (PI), S Leach, Prof. J Sharma, R Brown, C Taylor, I Wahishi, S Arif, S Bell, A Fields, S Butler, J Hindle, E Watson, J Borley, C Hewitt); University Hospital Aintree, Liverpool [60] (C Cullen (PI), D Hamill, Z Mellor, T Fluskey, V Hankin, A Keeling, R Durairaj, D Wood, J Peters, D Shackcloth, R Tangney, T Hlaing, V Sutton, M Harrison, S Stevenson, J Ewing); Bradford Royal Infirmary, Bradford [59] (C Patterson (PI), J Price, H Wilson, H Ramadan, S Maguire, S Khan, R Bellfield, U Hamid, M Hooley, R Ghulam, L Masters, W Gaba, O Quinn); Luton & Dunstable NHSFT University Hospital, Luton [56] (L Sekaran (PI), M Tate, N Mohammed, S Sethuraman, L Alwis, R Robinson, K Bharaj, R Pattni, F Justin, C Tam, M Chauhan, L Eldridge, S Mintias, J Palmones); Bristol Royal Infirmary, Bristol [54] (C Holmes (PI), L Guthrie, P Murphy, N Devitt, J Leonard, M Osborn, L Ball, A Steele, E Dodd, A Holloway, P Baker, R Patel, I Penwarden, S Caine, S Clarke, L Dow, S Williams, R Wynn-Williams); John Radcliffe Hospital, Oxford [51] (J Kennedy (PI), A DeVeciana, P Mathieson, I Reckless, R Teal, U Schulz, Prof. G Ford, P Mccann); St Georges Healthcare NHS Trust, London [47] (G Cluckie (PI), G Howell, J Ayer, B Moynihan, R Ghatala, B Clarke, G Cloud, B Patel, U Khan, N Al-Samarrai, F Watson, T Adedoyin, S Trippier, N Chopra, L Zhang, L Choy, K Kennedy, R Williams, V Jones, N Clarke, A Dainty, A Blight); South Glasgow University Hospital, Glasgow [45] (J Selvarajah (PI), W Smith, F Moreton, A Welch, D Kalladka, B Cheripelli, E Douglas, A Lush, X Huang, S El Tawil, N Day, K Montgomery, H Hamilton, D Ritchie, S Ramachandra, K McLeish); Northwick Park Hospital, Harrow [44] (D Cohen (PI), B Badiani, M Abdul-Saheb, A Chamberlain, M Mpelembue, R Bathula, M Lang, J Devine, L Alwis, L Southworth, L Burgess, N Epie, A David, E Owoyele, F Guo, A Oshodi, V Sudkeo); Royal Bournemouth Hospital, Bournemouth [44] (K Thavanesan (PI), D Tiwari, J Bell, C Ovington, E Rogers, R Bower, G Hann, B Longland, O David, A Hogan, S Loganathan, J Roberts, C Cox, S Orr, M Keltos); Yeovil District Hospital, Yeovil [41] (K Rashed (PI), D Wood, B Williams-Yesson, J Board, S De Bruijn, C Buckley, C Vickers, S Board, J Allison, E Keeling, T Duckett, D Donaldson, C Barron, L Balian, A Edwards, J Wilson); Royal Derby Hospital, Derby [40] (T England (PI), A Hedstrom, E Bedford, M Harper, E Melikyan, W Abbott, K Subramanian, M Goldsworthy); The Princess Royal Hospital, Telford [40] (M Srinivasan (PI), I Mukherjee, U Ghani, A Yeomans, D Donaldson, F Hurford, R Chapman, S Shahzad, O David, N Motherwell, L Tonks, R Young); Gloucestershire Royal Hospital, Gloucester [39] (D Dutta (PI), P Brown, F Davis, D Ward, J Turfrey, M Obaid, B Cartwright, B Topia, J Spurway, C Hughes, L Hill, S OConnell, K Collins, R Bakawala); Countess of Chester Hospital, Chester [38] (K Chatterjee (PI), T Webster, S Haider, P Rushworth, F Macleod, C Perkins, A Nallasivan, E Burns, S Leason, T Carter, S Seagrave); Airedale General Hospital, Keighley [37] (E Sami (PI), S Parkinson, M Hassan, S Naqvi, L Armstrong, S Mawer, G Darnbrook, C Booth, B Hairsine, M Smith, S Williamson, F Farquhar); Queen Elizabeth Hospital, Gateshead [36] (B Esisi (PI), T Cassidy, B McClelland, G Mankin, M Bokhari, D Sproates); Walsall Manor Hospital, Walsall [35] (E Epstein (PI), R Blackburn, S Hurdowar, N Sukhdeep, S Razak, N Upton, A Hashmi, K Osman); New Cross Hospital, Wolverhampton [34] (K Fotherby (PI), A Willberry, D Morgan, G Sahota, K Jennings-Preece, D Butler, S Das, A Stevens, N Ahmad, K Kauldhar); Royal Cornwall Hospital, Truro [34] (F Harrington (PI), A Mate, J Skewes,

K Adie, K Bond, G Courtauld, C Schofield, L Lucas, A James, S Ellis, B Maund, L Allsop, C Brodie, M Johnson, E Driver, K Harris, M Drake, K Moore, E Thomas); Wycombe Hospital, High Wycombe [34] (M Burn (PI), A Hamilton, S Mahalingam, A Benford, D Hilton, F Reid, A Misra, L Hazell, K Ofori, M Mathew, A Thomas, S Dayal, I Burn); University Hospital North Durham, Durham [32] (D Bruce (PI), M Naeem, R Burnip, R Hayman, P Earnshaw, E Brown, S Clayton, P Gamble, S Dima, M Dhakal, G Rogers, L Stephenson, R Nendick, Y Pai, K Nyo); Victoria Hospital, Kirkcaldy [32] (V Cvoro (PI), M Couser, M Simpson, A Tachtatzis, K Ullah, K McCormick, R Cain, N Chapman, S Pound, S McAuley); William Harvey Hospital, Ashford [32] (D Hargroves (PI), B Ransom, K Mears, K Griffiths, L Cowie, T Hammond, T Webb, I Balogun, H Rudenko, A Thomson, D Ceccarelli, A Gillian, E Beranova, A Verrion, N Chattha, N Schumacher, A Bahk, S Walker); Queen Elizabeth Hospital, Birmingham [31] (D Sims (PI), R Jones, J Smith, R Tongue, M Willmot, C Sutton, E Littleton, J Khaira, S Maiden, J Cunningham, Y Chin, C Green, M Bates, K Ahlquist); Royal Sussex County Hospital, Brighton [31] (I Kane (PI), J Breeds, T Sargent, L Latter, A Pitt Ford, T Levett, N Gainsborough, P Thompson, A Dunne, E Barbon, S Hervey); Poole Hospital, Poole [30] (S Ragab (PI), T Sandell, C Dickson, S Power, J Dube, N Evans, B Wadams, S Elitova, B Aubrey, T Garcia); Victoria Hospital, Blackpool [29] (J Mcilmoyle (PI), A Ahmed, C Dickinson, C Jeffs, S Dhar, K Jones, J Howard, C Armer, J Frudd, S Kumar, A Potter, S Donaldson); Watford General Hospital, Watford [29] (D Collas (PI), S Sundayi, L Denham, D Oza, E Walker, J Cunningham, M Bhandari); Sandwell General Hospital, Birmingham [28] (S Ispoglou (PI), R Evans, K Sharobeem, A Hayes, J Howard-Brown, E Walton, S Shanu, S Billingham); Southampton General Hospital, Southampton [27] (N Weir (PI), G Howard, E Wood, L Sykes, V Pressly, P Crawford, H Burton, A Walters, J Marigold, R Said, C Allen, S Evans, S Egerton, J Hakkak, J Andrews, R Lampard, S Smith, C Cox, S Tsang, R Creeden, I Gartrell, F Smith); The County Hospital, Hereford [26] (C Jenkins (PI), F Price, J Pryor, A Hedges, L Moseley, L Mercer, C Hughes); Addenbrookes Hospital, Cambridge [25] (E Warburton (PI), D Handley, S Finlay, N Hannon, A Espanol, S Kelly, J Mcgee, Prof. H Markus, D Chandrasena, J Sesay, D Hayden, H Hayhoe, J Macdonald, M Bolton, J Mitchell, C Farron, E Amis, D Day, A Culbert, L Whitehead, S Crisp, J Francis); Sunderland Royal Hospital, Sunderland [25] (J OConnell (PI), E Osborne, R Beard, P Corrigan, A Smith, M Edwards, L Mokoena, N Sattar, M Myint, R Krishnamurthy); West Suffolk Hospital, Bury St Edmonds [25] (A Azim (PI), S Whitworth, A Nicolson, S Alam, J White, M Krasinska-Chavez, J Imam, S Chaplin, D Singh, J Curtis, L Wood); Western General Hospital, Edinburgh [25] (Prof M Dennis (PI), R Buchan, W Rutherford, J Skwarski, D McGowan); Forth Valley Royal Hospital, Larbert [24] (A Byrne (PI), C McGhee, A Smart, Prof. M MacLeod, F Donaldson, J Blackburn, C Copeland, J Wilson, R Scott); Royal Liverpool University Hospital, Liverpool [24] (P Fitzsimmons (PI), G Fletcher, A Manoj, P Cox, L Trainor, P Lopez, M Wilkinson, L Denny, K Kavanagh, H Allsop); Queen Alexandra Hospital, Portsmouth [23] (U Sukys (PI), S Valentine, D Jarrett, K Dodsworth, M Wands, C Watkinson, W Golding, N Khan, J Tandy, R Butler, K Yip, C James, Y Davies, M Williams, A Suttling); Royal Berkshire NHS Foundation Trust, Reading [23] (K Nagaratnam (PI), N Mannava, N Haque, N Shields, K Preston, G Mason, K Short, G Uitenbosch, G Lumsdale); Royal Preston Hospital, Preston [23] (H Emsley (PI), S Sultan, B Walmsley, S Ahmed, D Doyle, A McLoughlin, L Hough, B Gregary, S Raj); Wythenshawe Hospital, Manchester [22] (A Maney (PI), S Blane, G Gamble, A Hague, B Charles, B Duran, C Lambert, K Stagg); Musgrove Park Hospital, Taunton [21] (R Whiting (PI), S Brown, M Hussain, M Harvey, J Homan, L Foote, L Graham, C Lane, L Kemp, J Rowe, H Durman, L Brotherton, N Hunt, J Foot, A Whitcher, C Pawley); Norfolk and Norwich University Hospital, Norwich [21] (P Sutton (PI), S Mcdonald, D Pak, A Wiltshire, J Balami, C Self, J Jagger, A Metcalf, G Healey, M Crofts, A Chakrabarti, C Hmu, J Keshet-Price, G Ravenhill, C Grimmer, T Soe, I Potter, P Tam, M Langley); Aberdeen Royal Infirmary, Aberdeen [20] (M MacLeod (PI), P Cooper, M Christie, J Irvine, A Joyson, F Annison, D Christie, C Meneses, A Johnson, S Nelson, V Taylor, J Furnace, H Gow, J Reid, R Clarke); East Surrey Hospital, Redhill [19] (Y Abousleiman (PI), S Bloom, S Goshawk, J Purcell, T Beadling, S Collins, S Jones, S Sangaralingham, E Munuswamy Vaiyapuri, M Landicho, Y Begum, S Mutton, J Allen, J Lowe, M Hughes); The Royal Victoria Hospital, Belfast [19] (I Wiggam (PI), S Tauro, S Cuddy, B Wells); Derriford Hospital, Plymouth [17] (A Mohd Nor (PI), C Eglinton, N Persad, M Kalita, M Weinling, S Weatherby, D Lashley, A Pace, C Brown, A Mucha, A Shah, J Baker, M Marner, J Westcott, N Wilmshurst, R Cowan, D Waugh); Doncaster Royal Infirmary, Doncaster [17] (D Chadha (PI), M Fairweather, D Walstow, R Fong); Morriston Hospital, Swansea [17] (M Krishnan (PI), H Thompson Jones, C Lynda, C Hughes, C Clements, R Williams, T Anjum, S Sharon, D Lynne, L Thomas, S Tucker, D Colwill, P Jones); The Hillingdon Hospital NHS Foundation Trust, Uxbridge [17] (E Vasileiadis (PI), A Parry, C Mason, M Holden, K Petrides, T Nishiyama, H Mehta, S Mumani); Perth Royal Infirmary, Perth [16] (S Johnston (PI), C Almadenboyle, S Carson, S Ross, P Nair, M Stirling, E Tenbruck); James Cook University Hospital, Cleveland [15] (D Broughton (PI), A Annamalai, J Wong, D Tryambake, L Dixon, A Skotnicka, J Thompson, A Sigsworth, S Whitehouse, J Pagan); Lister Hospital,

Stevenage [15] (A Pusalkar (PI), H Beadle, K Chan, P Dangri, A Asokanathan, A Rana, S Gohil, K Crabtree, A Cook, M Massyn, P Aruldoss, S Dabbagh); Salisbury District Hospital, Salisbury [15] (T Black (PI), C Clarke, R Fennelly, L Nardone, V DiMartino, A Anthony, D Mead, M Tribbeck); St Peters Hospital, Chertsey [14] (B Affley (PI), C Sunderland, E Young, L Goldenberg, A Khan, P Wilkinson, L Abbott, R Nari, S Lock, J Stewart, A Shakhon, R Pereira, M DSouza, S Dunn, N Cron, A Mckenna); Colchester General Hospital, Colchester [13] (R Sivakumar (PI), S Cook, A Wright, J Ngeh, R Saksena, J Ketley-O'Donel, R Needle, E Chinery); University College London Hospitals NHS Foundation Trust, London [13] (R Greenwood (PI), L Howaniec, C Watchurst, K Patel, R Erande, M Brezitski, N Passeron, E Elliott, N Oji, D Austin, A Banaras, C Hogan, T Corbett, R Shah); Warrington Hospital, Warrington [13] (M Kidd (PI), G Hull, J Simpson, S Punekar, J Nevinson, H Penney, J Ward, W Wareing, N Hayes, K Bunworth, L Connell, K Mahawish, G Drummond); Worthing Hospital, Worthing [13] (N Sengupta (PI), M Metiu, C Gonzalez, J Margalef, S Funnell, G Peters, I Chadbourn); Dorset County Hospital, Dorchester [12] (H Proeschel (PI), P Ashcroft, S Sharpe, S Jones, P Cook, D Jenkinson, D Kelly, H Bray); Queen Elizabeth The Queen Mother Hospital, Margate [12] (G Gunathilagan (PI), K Griffiths, K Mears, A Gillian, S Jones, S Tilbey, S Abubakar, E Beranova); Kings Mill Hospital, Mansfield [11] (M Cooper (PI), A Rajapakse, A Nasar, J Janbieh, L Wade, L Otter, I Wynter, S Haigh, R Boulton, J Burgoyne, A Boulton); Stepping Hill Hospital, Stockport [11] (J Vassallo (PI), A Hasan, L Orrell, A Khan, S Qamar, S Graham, D Leonard, E Hewitt, M Haque, J Awolesi, E Bradshaw, A Kent); Bronglais General Hospital, Aberystwyth [10] (P Jones (PI), C Duggan, A Hynes, E Nurse, S Raza, U Pallikona, B Edwards, G Morgan, H Tench, R Loosley, K Dennett, T Trugeon-Smith, R Jones, S Jones, R Williams, D Robson); Hull Royal Infirmary, Hull [10] (R Rayessa (PI), A Abdul-Hamid, V Lowthorpe, K Mitchelson, E Clarkson, H Rhian, A Fleming); Broomfield Hospital, Chelmsford [9] (R Kirthivasan (PI), J Topliffe, R Keskeys, S Williams, F McNeela, E Bohannan, L Cooper, S Shah, G Zachariah, F Cairns, T James, L Fergey, S Smolen, A Lyle, E Cannon, S Omer); Whiston Hospital, Prescot [9] (S Mavinamane (PI), S Meenakshisundaram, L Ranga, J Bate, A Hill, M Hargreaves, T Smith, S Dealing, L Harrison); Frimley Park Hospital, Frimley [8] (S Amlani (PI), G Gulli, M Hawkes-Blackburn, N Hunter, S Levy, L Francis, S Holland, A Peacocke, J Amero, M Burova, O Speirs); Harrogate Hospital, Harrogate [7] (S Brotheridge (PI), S Al Hussayni, H Lyon, C Hare, S Jackson, L Stephenson, J Featherstone, A Bwalya); Royal Blackburn Hospital, Blackburn [7] (A Singh (PI), M Goorah, J Walford, A Bell, C Kelly, D Rusk, D Sutton, F Patel, S Duberley, K Hayes, L Hunt); Scarborough Hospital, Scarborough [7] (E Ahmed El Nour (PI), S Dyer, L Brown, K Elliott, E Temlett, J Paterson, P Wood, M Haritakis, S Honour, C Box, P Cottrell, J Westmoreland, S Young, R Furness); West Cumberland Hospital, Whitehaven [7] (E Orugun (PI), H Crowther, R Glover, C Brewer, S Thornthwaite); Macclesfield District General Hospital, Macclesfield [6] (M Sein (PI), K Haque, L Bailey, S Wong, E Gibson, K Burton, L Brookes, K Rotchell, K Waltho, C Lindley, A Murray, D Leonard, M Holland); Royal Lancaster Infirmary, Lancaster [6] (P Kumar (PI), M Khan, P Harlekar, c culmsee, L Booth, J Drew, J Ritchie, N Mackenzie, C Thomas, J Barker); Weston General Hospital, Weston Super Mare [6] (M Haley (PI), D Cotterill, L Lane, D Simmons, R Warinton, G Saunders, H Dymond, S Kidd, C Little, Y Neves-Silva); Basildon and Thurrock University Hospital, Basildon [5] (B Nevajda (PI), M Villaruel, S Patel, U Umasankar, A Man, N Gadi, N Christmas, R Ladner, R Rangasamy, G Butt, W Alvares); Ulster Hospital, Belfast [5] (M Power (PI), S Hagan, K Dynan, D Wilson, S Crothers, C Leonard, B Wroath, G Douris); Antrim Area Hospital, Antrim [4] (D Vahidassr (PI), B Gallen, S McKenna, A Thompson, C Edwards, C McGoldrick, M Bhattad); Epsom General Hospital, Epsom [3] (J Putteril (PI), R Gallifent, E Makanju, M Lepore, C McRedmond, L Arundell, A Goulding); Fairfield General Hospital, Bury [3] (K Kawafi (PI), P Jacob, L Turner, N Saravanan, L Johnson, D Morse, R Namushi, S Humphrey, R Patel, J McLaughlin); Leighton Hospital, Crewe [3] (M Salehin (PI), S Tinsley, T Jones, D Bailey, L Garcia-Alen, L Kalathil, R Miller, N Gautam, J Horton, J Meir, E Margerum, A Ritchings, A Jones, K Amor); Royal Free Hospital, London [3] (V Nadarajan (PI), J Laurence, S Fung Lo, S Melander, P Nicholas, E Woodford, G McKenzie, V Le, J Crause); St Helier Hospital, Carshalton [3] (P OMahony (PI), C Orefo, C McDonald, V Jones, E Osikominu, T Khan, G Appiatse, E Makanju, A Wardale, M Augustin, H Stone); North Middlesex University Hospital NHS Trust, London [2] (R Luder (PI), M Bhargava, G Bhome, V Johnson, R Shah, D Chesser, H Bridger, E Murali); South Tyneside General Hospital, South Shields [2] (J Scott (PI), S Morrison, A Burns, J Graham, M Duffy); Princess Royal Hospital, Haywards Heath [1] (K Ali (PI), T Sargent, E Pitcher, J Gaylard, J Newman);

Rotherham Hospital, Rotherham [1] (S Punnoose (PI), M Khan, S Oakley, V Murray, C Bent, R Walker, K Purohit, A Rees, M Davy, S Besley, O Chohan); Royal London Hospital (Barts Health), London [1] (L Argandona (PI), L Cuenoud, H Hassan, E Erumere, A OCallaghan, L Howaniec, O Redjep, G Auld, P Gompertz, A Song, R Hungwe, H Kabash, T Tarkas); Royal Surrey County Hospital, Guildford [1] (A Blight (PI), S Jones, G Livingstone, F Butler, S Bradfield, L Gordon, J Schmit, A Wijewardane, C Medcalf, T Edmunds, R Wills, C Peixoto).

**Contributors** GM and MD were co-principal investigators of the Fluoxetine Or Control Under Supervision (FOCUS) trial and co-led all aspects of the trial from inception to completion and dissemination. Please note that the author is the FOCUS trial collaboration, and GM and MD are the writing committee. GM is the guarantor.

**Funding** This work was supported by Stroke Association (grant number TSA 2011101) and National Institute for Health Research (NIHR) Health Technology Assessment Programme (project number 13/04/30).

**Disclaimer** The views and opinions expressed herein are those of the authors and do not necessarily reflect those of the NIHR Health Technology Assessment Programme.

**Competing interests** None declared.

**Patient consent for publication** Not required.

**Ethics approval** Scotland A Research Ethics Committee approved the trial and our approach, including our plans to inform participants of results.

**Provenance and peer review** Not commissioned; externally peer-reviewed.

**Data availability statement** Data are available upon reasonable request.

**ORCID iD**
Gillian Mead http://orcid.org/0000-0001-7494-2023

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
