## [Reviewer comments · BMJ Open]

ARTICLE DETAILS

TITLE (PROVISIONAL)	Feasibility of reporting results of large randomised controlled trials to participants: experience from the Fluoxetine or Control under supervision (FOCUS) trial
AUTHORS	Mead, Gillian; Dennis, Martin

VERSION 1 – REVIEW

REVIEWER	Eoin Dinneen University College London, UK.
REVIEW RETURNED	11-Jun-2020

GENERAL COMMENTS	The authors give an account of the manner in which they informed their trial participants of the trial results and the individuals' treatment allocation at the end of the study, co-inciding with the publication of the study in the Lancet in December 2018. The article is well-written and thought-provoking. Abstract is clear and concise. I have a couple of questions, which if addressed I think would improve the article and, which I think could be easily addressed by the authors. a) the trial team took the email address of the participant at the 6-mo and 12-mo time point, but was email address and contact information not collected at recruitment to the trial? b) page 6 line 45. 'considering whether to disseminate the results to all participants by post but this would have required substantial resource'. are the authors able to give more in terms of the cost of this? This is interesting for 3 reasons, i) the purpose of this article is to discuss feasibility of this process, which often includes costings of a process/intervention, ii) it would be of interest to other researchers considering how to contact their trial participants, iii) the NIHR HTA grant for the study included a 'request for funding for dissemination of results that was approved' - so how did the cost match up with the funding that was provided for this endeavour. c) out of interest (page 6 line 11) how was the timing of the publication of the paper known such that the email could be sent at exactly the same time on the 5th December? d) a copy of the email sent to participants might make a nice supplementary material. How was their allocation revealed? how was the results of the trial given to patients (abstract, lay summary,
--

	full study, and in what detail)? and what support mechanisms were offered? e) small question: 3 people called the telephone line. i) how does this compare to the number of participants who were calling the telephone line during the actual running of the trial? ii) if the investigators were to do it again, might they not re-open the telephone line as this might save on cost without removing a very well-used resource? Discussion is brief and relevant. Although not to have a single citation in the discussion is unusual. Notwithstanding that this is the first RCT to report informing participants of results and trial allocation, is there any supporting literature about this area or the merits and demerits of unblinding patients that might be called upon to contextualise this interesting brief report. Thank you.
--	---

REVIEWER	Christine Roffe University Hospital of North Midlands NHS Trust Stoke-on-Trent UK
REVIEW RETURNED	12-Jul-2020

GENERAL COMMENTS	Sharing results of clinical research with participants is considered best practice, but very rarely done. To my knowledge this is the only study where all participants who were contactable were given trial results as well as details of their allocated treatment at the same time as the publication of the trial results. The authors demonstrate that most participants or their next of kin want to be informed, that giving such information is feasible, and can be provided with little additional cost. It is reassuring that contacting families of deceased patients did not cause distress, and that the number of queries generated was manageable, and arrived within 2 weeks of receipt of the results. This is important work, transferable beyond stroke, and will guide information sharing for future clinical trials. I would like the authors to check numbers for the proportion of participants who did not wish to be informed. The stated 50% seems an overestimation. 1895 gave an email/address, implying they wanted to be informed. This is 60% of all participants and 70% of those who could be contacted at 12 months.
---

VERSION 1 – AUTHOR RESPONSE

Thank you for the referee comments which were most helpful. We have revised the manuscript accordingly

1. We have removed the Summary boxes as these are not required.
2. Reviewer 1 asked: ‘The trial team took the email address of the participant at the 6-mo and 12-mo time point, but was email address and contact information not collected at recruitment to the trial?’

Our response-We did not collect the email address at the time of recruitment. We clarified this (page 5)

3. Reviewer 1 asks about costing for disseminating results by post.

Our response: We have explained the estimated cost (page 5) and discussed how this matched up with the funding provided (£1 per letter).

4. Reviewer 1 asks how the timing of the publication of the paper known such that the email could be sent at exactly the same time on the 5th December?

Our response: we have explained this (page 5-6)

5. Review 1 states that a copy of the email sent to participants might make a nice supplementary material, and asks: How was their allocation revealed? how was the results of the trial given to patients (abstract, lay summary, full study, and in what detail)? and what support mechanisms were offered?

Our response. We agree and have included a copy of the email. We explained that it was written in lay terms.

6. Reviewer 1 asks: 3 people called the telephone line. i) how does this compare to the number of participants who were calling the telephone line during the actual running of the trial? ii) if the investigators were to do it again, might they not re-open the telephone line as this might save on cost without removing a very well-used resource?

Our response: These are important points. We have included responses in the discussion.

7. Reviewer 1: Discussion is brief and relevant. Although not to have a single citation in the discussion is unusual. Notwithstanding that this is the first RCT to report informing participants of results and trial allocation, is there any supporting literature about this area or the merits and demerits of unblinding patients that might be called upon to contextualise this interesting brief report.

Our response: We have searched the literature again (google scholar, using phrase 'informing participants of trial results) and have included a citation to a narrative review published in 2008. We found no other large RCTs since then that have reported how they disseminated results.

8. Reviewer 2 states: Sharing results of clinical research with participants is considered best practice, but very rarely done. To my knowledge this is the only study were all participants who were contactable were given trial results as well as details of their allocated treatment at the same time as the publication of the trial results. The authors demonstrate that most participants or their next of kin want to be informed, that giving such information is feasible, and can be provided with little additional cost. It is reassuring that contacting families of deceased patients did not cause distress, and that the number of queries generated was manageable, and arrived within 2 weeks of receipt of

the results. This is important work, transferable beyond stroke, and will guide information sharing for future clinical trials.

I would like the authors to check numbers for the proportion of participants who did not wish to be informed. The stated 50% seems an overestimation. 1895 gave an email/address, implying they wanted to be informed. This is 60% of all participants and 70% of those who could be contacted at 12 months.

Our response: We have checked the percentage. We had stated 60% of all participants in the abstract; and we have clarified that this is a proportion of the number randomised.

I hope that these changes are satisfactory, and I would be happy to clarify any further points.

Kind regards
Yours sincerely
Professor Gillian Mead